# APTIMA mRNA vs. DNA-Based HPV Assays: Analytical Performance Insights from a Resource-Limited South African Setting

**DOI:** 10.3390/ijms26157450

**Published:** 2025-08-01

**Authors:** Varsetile Varster Nkwinika, Kelvin Amoh Amissah, Johnny Nare Rakgole, Moshawa Calvin Khaba, Cliff Abdul Magwira, Ramokone Lisbeth Lebelo

**Affiliations:** 1Department of Virological Pathology, Sefako Makgatho Health Sciences University, Medunsa, P.O. Box 60, Pretoria 0204, South Africa; varsetile.nkwinika@smu.ac.za (V.V.N.); 201805336@swave.smu.ac.za (K.A.A.); nare.rakgole@smu.ac.za (J.N.R.); cmagwira@gmail.com (C.A.M.); 2South African Vaccination and Immunisation Centre (SAVIC), Sefako Makgatho Health Sciences University, Medunsa, P.O. Box 60, Pretoria 0204, South Africa; 3Department of Anatomical Pathology, Sefako Makgatho Health Sciences University, Medunsa, P.O. Box 60, Pretoria 0204, South Africa; moshawa.khaba@nhls.ac.za; 4National Health Laboratory Service (NHLS), Department of Anatomical Pathology, Sefako Makgatho Health Sciences University, Medunsa, P.O. Box 60, Pretoria 0204, South Africa; 5National Health Laboratory Service (NHLS), Department of Virological Pathology, Sefako Makgatho Health Sciences University, Medunsa, P.O. Box 60, Pretoria 0204, South Africa

**Keywords:** E6/E7 mRNA, HPV DNA, cervical cancer screening, analytical performance, South Africa

## Abstract

Cervical cancer remains a major health burden among women in sub-Saharan Africa, where screening is often limited. Persistent high-risk human papillomavirus (HR-HPV) infection is the principal cause, highlighting the need for accurate molecular diagnostics. This cross-sectional study evaluated the analytical performance of one mRNA assay, APTIMA^®^ HPV assay (APTIMA mRNA), and two DNA-based assays, the Abbott RealTime High Risk HPV assay (Abbott DNA) and Seegene Allplex™ II HPV28 assay (Seegene DNA), in 527 cervical samples from a South African tertiary hospital, focusing on 14 shared HR-HPV genotypes. Seegene DNA yielded the highest detection rate (53.7%), followed by Abbott DNA (48.2%) and APTIMA mRNA (45.2%). APTIMA mRNA showed a strong agreement with Abbott DNA (87.9%, κ = 0.80), 89.9% sensitivity, 91.2% NPV, and the highest accuracy (AUC = 0.8804 vs. 0.8681). The agreement between APTIMA mRNA and Seegene DNA was moderate (83.4%, κ = 0.70), reflecting target differences. Many DNA-positive/mRNA-negative cases likely represent transient infections, though some may be latent with reactivation potential, warranting a follow-up. In resource-constrained settings, prioritizing transcriptionally active infections through mRNA testing may enhance screening efficiency and reduce burden. Scalable, cost-effective assays with strong clinical utility are essential for broadening access and improving cervical cancer prevention. Further studies should assess the integration of mRNA testing into longitudinal screening algorithms.

## 1. Introduction

Persistent infection with high-risk human papillomavirus (HR-HPV) is the primary cause of cervical cancer, with HPV types 16 and 18 being the most frequently detected [1,2]. In response to the limitations (poor sensitivity) of cytology-based screening, molecular tests that detect HPV DNA or mRNA, are increasingly used for primary screening due to their higher clinical sensitivity for identifying women at risk of high-grade cervical lesions (CIN2/3) [3,4,5].

Among the molecular tests, mRNA-based assays provide a key clinical advantage by detecting E6/E7 oncogene transcripts, which are indicators of transcriptionally active and potentially transforming HPV infections [5,6,7]. This improves diagnostic specificity and risk stratification compared to DNA-based assays, which may detect transient and clinically insignificant infections [5,8,9,10]. However, mRNA assays have shown slightly lower analytical sensitivity compared to DNA-based platforms [8,11], and comparative studies, such as that by Dockter et al. [11], have highlighted this trade-off when evaluating mRNA-based detection against widely used DNA assays, like Qiagen’s Hybrid Capture 2 test.

DNA-based assays, such as the Abbott RealTime High Risk HPV (hereafter Abbott DNA), Hybrid Capture 2, and Roche Cobas 4800 are widely validated and commonly used [9,10]. These platforms typically target conserved regions of the viral genome, often reporting HPV 16 and 18 separately due to their high oncogenic risk [8,9,10]. However, in high-burden, resource-limited settings, their limited specificity may result in over-referral and overtreatment [10,12].

Furthermore, as HPV vaccination coverage expands, shifts in genotype prevalence, including possible type replacement or unmasking of non-vaccine types, necessitate ongoing molecular surveillance [8,12]. This is particularly critical in sub-Saharan Africa, where HPV types, such as 33, 35, 45, 52, and 58, are common [12,13,14], with HPV 35 sometimes competing with HPV 16 in cervical cancer cases [14,15,16].

To support such surveillance and improve screening, extended genotyping platforms are increasingly used [10]. The Seegene Allplex™ II HPV28 assay (hereafter Seegene DNA) enables the detection of 28 HPV genotypes via multiplex real-time PCR, offering high-throughput processing, automation, and strong agreement with FDA-approved assays [8,17].

Despite these advancements, the inability of DNA assays to differentiate between transient and transforming infections limits their clinical performance [11,18]. mRNA assays, such as the APTIMA^®^ HPV assay (hereafter APTIMA mRNA), have emerged as more specific tools for identifying clinically significant infections [6,7,9]. Large trials like ESTAMPA have reinforced the value of molecular screening in diverse populations [19], and the World Health Organization (WHO) now recommends a global shift to HPV-based screening, particularly in resource-limited settings [20].

This study assessed the screening performance of the APTIMA mRNA with two DNA-based platforms, the Seegene DNA and Abbott DNA assays, in cervical samples from a South African cohort. Previous studies, including a screen-and-treat trial by Sørbye et al. [21], have demonstrated the utility of genotype-specific HPV E6/E7 mRNA testing in improving screening specificity and reducing overtreatment in South African women. However, while comparisons between mRNA and DNA-based assays (such as Cobas or Abbott DNA) have been reported [8,10,11,22,23], there remains a lack of published evidence directly comparing mRNA assays with extended genotyping platforms like the Seegene DNA. This study addresses that gap by comparing the analytical performance of the Abbott DNA, Seegene DNA, and APTIMA mRNA assays, evaluating their relative sensitivity and specificity for detecting 14 HR-HPV types. By including an extended genotyping assay, this study therefore contributes new comparative data to inform evidence-based screening strategies that improve diagnostic accuracy, reduce overtreatment, and support cervical cancer prevention in high-prevalence and resource-constrained settings.

## 2. Results

### 2.1. Analytical Performance of mRNA vs. DNA-Based HPV Assays

To compare analytical performance and inter-assay agreement, each of the three assays, APTIMA mRNA, Seegene DNA, and Abbott DNA, was sequentially used as the reference standard against which the others were evaluated (Table 1, Table 2 and Table 3). Although the Seegene DNA assay detects 28 HPV types, the analysis was restricted to the 14 HR-HPV types common to all three platforms to ensure comparability. Among the 527 samples tested, HR-HPV prevalence was 48.2% with Abbott DNA (Table 1a,b), 53.7% with Seegene DNA (Table 2a,b), and 45.2% with APTIMA mRNA (Table 3a,b). These proportions form the basis for the comparative performance assessment.

When Abbott DNA was used as the reference standard (Table 1a,b), Seegene DNA demonstrated higher specificity at 91.4% (95% CI: 87.1–94.6) compared to APTIMA mRNA’s 86.2% (95% CI: 81.6–89.9), indicating Seegene DNA classified Abbott-negative samples as negative more frequently. Seegene DNA also had a higher positive predictive value (PPV) of 91.7% (95% CI: 87.6–94.9) versus 84.3% (95% CI: 79.2–88.5) for APTIMA mRNA, reflecting a higher proportion of positive Seegene DNA results confirmed by Abbott DNA. However, Seegene DNA showed lower sensitivity at 82.3% (95% CI: 77.3–86.5) and negative predictive value (NPV) of 81.6% (95% CI: 76.4–86.1) compared to APTIMA mRNA’s higher sensitivity of 89.9% (95% CI: 85.4–93.4) and NPV of 91.2% (95% CI: 87.2–94.3), indicating APTIMA mRNA detected a higher proportion of Abbott-positive samples and correctly classified Abbott-negative samples as negative. Seegene DNA’s overall agreement with Abbott DNA was 82.9% and Cohen’s kappa was 0.7, both lower than APTIMA mRNA’s agreement of 87.9% and kappa of 0.8, indicating stronger concordance of APTIMA mRNA with the Abbott DNA. Although APTIMA mRNA’s specificity was slightly lower than Seegene DNA’s, this may reflect its focus on detecting transcriptionally active infections via mRNA, whereas DNA assays detect both active and transient infections.

When Seegene DNA was used as the reference standard (Table 2a,b), Abbott DNA demonstrated a slightly higher overall agreement (86.5% vs. 83.4%), sensitivity (91.7%, 95% CI: 87.6–94.8 vs. 91.1%, 95% CI: 86.8–94.4), specificity (81.6%, 95% CI: 76.5–86.0 vs. 77.1%, 95% CI: 71.8–81.8), and PPV (82.3%, 95% CI: 77.3–86.5 vs. 76.6%, 95% CI: 71.2–81.4) compared to APTIMA mRNA. The NPV was similar between Abbott DNA and APTIMA mRNA (91.4%, 95% CI: 87.1–94.6 for both), as was Cohen’s kappa (*k* = 0.7 for both). These results suggest Abbott DNA aligns more closely with Seegene DNA in identifying both positive and negative cases, while APTIMA mRNA maintains a comparable performance in ruling out negatives but with slightly reduced concordance on positives.

On the other hand, when APTIMA mRNA was used as the reference standard (Table 3a,b), Abbott DNA demonstrated higher overall agreement (87.9% vs. 83.4%), sensitivity (84.3%, 95% CI: 79.2–88.5 vs. 76.6%, 95% CI: 71.2–81.4), NPV (86.2%, 95% CI: 81.6–89.9 vs. 77.1%, 95% CI: 71.8–81.8), and Cohen’s kappa (*k* = 0.8 vs. 0.7) results compared to Seegene DNA. In contrast, Seegene DNA showed higher specificity (91.4%, 95% CI: 87.1–94.6 vs. 86.2%, 95% CI: 81.6–89.9) and PPV (91.1%, 95% CI: 86.8–94.4 vs. 89.9%, 95% CI: 85.4–93.4) than Abbott DNA. This suggests that Abbott DNA was more closely aligned with APTIMA mRNA in detecting both positive and negative cases, whereas Seegene DNA was more consistent in confirming positive results. These findings highlight how assay performance varies depending on which test is used as the reference standard, particularly in terms of sensitivity and specificity.

#### Receiver Operating Characteristic (ROC) Curve Analyses

Figure 1 displays the ROC analyses using Abbott DNAas the reference test. Figure 1a compares APTIMA mRNA to Abbott DNA, showing an area under the curve (AUC) of 0.8398 for APTIMA mRNA and 0.8681 for Abbott DNA, indicating strong overall concordance. Figure 1b compares APTIMA mRNA and Seegene DNA assays. APTIMA mRNA showed a marginally higher AUC (0.8804) than Seegene DNA (0.8681), though the difference was not statistically significant (*p* = 0.0563). These ROC comparisons reflect analytical agreement and are not measures of clinical diagnostic accuracy, as no histologic reference standard (e.g., CIN2/3) was available.

### 2.2. Comparative Detection of HR-HPV Types Across HPV Assays

Table 4a–c present pairwise comparisons of 14 HR-HPV genotypes (HPV 16, 18, 31, 33, 35, 39, 45, 51, 52, 56, 58, 59, 66, and 68) detected by the APTIMA mRNA and 2 DNA-based assays: Abbott DNA and Seegene DNA. The differences in HR-HPV positivity rates among the assays have been described previously in Section 2.1 (Seegene DNA 53.7%, Abbott DNA 48.2%, and APTIMA mRNA 45.2%) and provide context for these comparisons. In the APTIMA mRNA versus Abbott DNA comparison (Table 4a), 4.6% of samples were APTIMA-positive but Abbott-negative. Similarly, in the APTIMA mRNA vs. Seegene DNA comparison (Table 4b), 4.0% were APTIMA-positive but Seegene-negative. These mRNA-positive/DNA-negative cases likely represent transcriptionally active infections with low or undetectable viral DNA, rather than false positives. Abbott and Seegene detected 7.6% and 12.6% DNA-positive/mRNA-negative samples, respectively, consistent with DNA assays’ higher sensitivity to detect latent or transient infections without active oncogene expression. The Abbott DNA versus Seegene DNA comparison (Table 4c) shows that 9.5% of samples are positive by Seegene DNA but negative by Abbott DNA, indicating some discrepancies between the DNA assays in detecting these genotypes. This difference may impact genotype surveillance and epidemiological monitoring by affecting the accurate detection and tracking of circulating HR-HPV types in the study population.

### 2.3. HPV Genotype Distribution and Implications for Vaccine Coverage

The overall genotype distribution detected by the Seegene DNA is shown in Figure 2. Among HR-HPV types, HPV16 was the most prevalent (27.9%) and HPV35 (20.0%) was the fourth most common HR type. Low-risk types were less frequent, with HPV40 (11.1%) being the most common among them. Notably, HPV6 and 11, targeted by current vaccines, were detected in few samples, suggesting regional variation or population-specific genotype distribution.

Figure 3 illustrates the distribution of HPV genotypes detected among the 315 HPV-positive individuals across all the tests, categorized according to coverage by currently licensed HPV vaccines. Based on the detected genotypes, Cervarix, which targets HPV16 and 18, would have provided protection to 36.2% of individuals. The inclusion of LR-types HPV6 and 11 in Gardasil extends this coverage slightly to 41.0%. Gardasil-9, which also targets 5 additional HR HPV types (HPV31, 33, 45, 52, and 58), would have increased the potential vaccine coverage to 73.0% of the cases.

### 2.4. Sociodemographic Predictors of HR-HPV Infection

Table 5 summarizes the HPV detection rates across the three assays, based on all genotypes detected by each platform, and examines their socio-demographic associations. The highest overall HPV prevalence was observed with the Seegene DNA (60.0%, 315/525), which detects all 28 genotypes included in this study. Abbott DNA and APTIMA mRNA assays, which detect only 14 HPV types, previously reported prevalence rates of Abbott DNA 48.2%, (254/527) and APTIMA mRNA 45.0% (237/527), respectively. Seegene DNA’s broader analytical coverage likely accounts for its higher detection rate, whereas APTIMA mRNA’s lower prevalence reflects its focus on transcriptionally active infections through mRNA detection.

Across all assays, HPV positivity was statistically associated with employment and marital/relationship status (*p* < 0.001). Women who were unemployed and single had notably higher infection rates. No statistically significant associations were found for age, ethnicity, province, or residential location, although there was a difference in number of participants by province and location where the majority were from Gauteng (90.2%) and Semi-Urban (86.8%) areas.

## 3. Discussion

HPV DNA-based testing has become a valuable tool in cervical cancer screening due to its high sensitivity compared to traditional cytology. However, its lower specificity poses a challenge because it cannot reliably differentiate between infections that will resolve on their own and those that persist and carry a higher risk of progressing to disease. Distinguishing persistent infections is crucial to minimize unnecessary worry for women who test positive and to avoid overwhelming healthcare systems with follow-ups and treatment for cases that are unlikely to advance. This concern is especially important in settings with limited resources, such as South Africa, where the careful allocation of healthcare services is essential. South Africa’s recent implementation of HPV testing in its national screening program highlights the importance of using diagnostic methods that balance accuracy and operational feasibility.

To support such context-sensitive screening strategies, this study compared the analytical performance and inter-assay agreement of three HPV assays, APTIMA mRNA, Seegene DNA, and Abbott DNA, using a sequential reference comparator design. The results demonstrate that performance varies depending on the reference assay used. Overall, Abbott DNA showed an intermediate agreement with both Seegene DNA (86.5%) and APTIMA mRNA (87.9%), indicating a consistent performance across platforms. Seegene DNA exhibited higher specificity (91.4%) and positive predictive value (PPV: ranging from 71.7% to 91.1%) across comparisons, suggesting a more stringent positivity threshold. Seegene DNA’s broader genotypic coverage likely contributed to its higher overall HPV positivity rate (60.0%) compared to APTIMA mRNA (45.2%), supporting its utility for surveillance and expanded genotyping purposes [8]. Nonetheless, as a DNA-based assay, Seegene DNA may still detect transient or clinically insignificant infections, limiting its ability to distinguish infections that are likely to resolve from those at risk of progression [8,22].

In contrast, APTIMA mRNA demonstrated marginally higher sensitivity (ranging from 89.9% to 91.1%) and negative predictive value (NPV: ranging from 91.2% to 91.4), attributable to its detection of E6/E7 mRNA transcripts, which are markers of transcriptionally active HPV infections with the highest oncogenic potential [6,7,11]. This enhances its clinical relevance, particularly in identifying persistent infections more likely to progress to cervical disease [24,25]. These findings highlight the importance of aligning assay selection with intended clinical or public health goals. While Seegene DNA may be preferable for epidemiological monitoring and vaccine impact assessment, APTIMA mRNA may be better suited for clinical triage where prioritizing actionable infections is critical [9,22,23,24,25,26]. Given the variability in disease burden, follow-up infrastructure, and screening capacity across LMICs, a one-size-fits-all approach may be suboptimal. Instead, integrating DNA- and mRNA-based assays within tailored screening strategies could help balance sensitivity, specificity, and sustainability in diverse health system contexts [21].

In resource-limited settings, mRNA-based HPV testing, such as APTIMA mRNA, offers clinical and programmatic advantages by identifying transcriptionally active infections with a higher likelihood of progression. This enables more precise triaging compared to DNA-based assays, which cannot differentiate between transient and persistent infections [10,11]. For example, consider a woman who tests positive on an HPV DNA assay but negative for mRNA. Rather than being referred for immediate colposcopy, a costly and invasive procedure with limited availability, she could be scheduled for repeat testing. This approach not only reduces patient anxiety and the risk of overtreatment but also conserves specialist services. mRNA testing therefore supports more rational clinical management, optimizing treatment resource allocation by prioritizing women who are most likely to benefit from further intervention. These features are advantageous within the framework of WHO’s screen-and-treat strategies, which prioritize the use of molecular tools that balance high specificity with programmatic efficiency [20,21].

Nonetheless, the implementation of mRNA-based platforms, like APTIMA mRNA, as primary screening tools requires careful consideration. Although APTIMA mRNA improves clinical specificity [11], it may require adjunctive strategies, such as HPV genotyping or reflex cytology, to optimize triage and ensure accurate case management. In LMICs, the financial and operational implications of using multiple high-cost assays cannot be overlooked. Therefore, there is an urgent need to develop affordable, scalable mRNA-based platforms that maintain biological relevance and can be feasibly integrated into national screening programs. The strategic focus should extend beyond commercial assays to scalable mRNA-based approaches that improve predictive value and support WHO’s vision of transitioning toward HPV-based screening tailored to local realities.

Receiver operating characteristic (ROC) curve analysis further supported APTIMA mRNA’s superior discriminatory capacity, with an area under the curve (AUC) of 0.8804 compared to 0.8681 for Seegene DNA (*p* = 0.0563). While not statistically significant, this difference suggests that APTIMA mRNA may better differentiate between potentially pathogenic and benign infections based on molecular signatures [19,24]. However, it is important to clarify that as cytological or histological endpoints were not included, the findings pertain to analytical rather than clinical performance.

Differences in assay performance likely reflect underlying methodological distinctions. For instance, Abbott and Seegene both target the L1 region using real-time PCR, but differ in approach, where Abbott employs TaqMan hydrolysis probes to detect 14 HR-HPV types, while Seegene uses multiplex PCR with Dual-Priming Oligonucleotide (DPO) primers and tagging oligonucleotide cleavage and extension (TOCE) technology to simultaneously genotype 28 HPV types [8,10]. In contrast, APTIMA mRNA targets E6/E7 mRNA transcripts, markers of viral oncogenic activity, using transcription-mediated amplification (TMA). These transcripts are typically upregulated in persistent infections, making mRNA detection a more biologically relevant approach for identifying infections with transformation potential [10,22]. However, while APTIMA mRNA targets the 14 most common and clinically relevant HR-HPV genotypes, it does not detect some probable HR types, such as HPV 26, 53, 73, and 82. In certain epidemiological contexts, particularly where these genotypes circulate more frequently, this limitation may reduce the assay’s diagnostic scope. Nonetheless, the 14 genotypes covered by APTIMA mRNA account for the majority of types associated with cervical cancer cases globally [2,5], reinforcing its practicality and value in high-burden populations requiring efficient, high-impact screening tools. Furthermore, it is important to note that the assays differ in their pre-analytical protocols, which may influence analytical performance. For instance, both DNA-based tests (Abbott DNA and Seegene DNA) were performed using the same nucleic acid extracts obtained from 0.8 mL of sample. In contrast, HPV mRNA testing (APTIMA mRNA) was performed using a different nucleic acid extraction protocol and starting from 1 mL of sample, which may contribute to variability in assay sensitivity and specificity.

This fundamental difference in biological targets complicates direct assay comparison. Nonetheless, comparing mRNA- and DNA-based assays remains relevant when assessing their suitability for screening, where both analytical characteristics and clinical implications must be carefully considered [10,11,21]. In the context of this study, APTIMA mRNA’s performance supports its potential role in programs aiming to prioritize the detection of persistent, high-risk infections over transient colonization, an approach that aligns with the overarching theme of improving strategies for detecting persistent HPV infections.

The genotype distribution derived from the Seegene Allplex™ HPV28 assay was dominated by HPV16, followed by HPV58, 66, 35, and 18, mirroring previously reported data in South African populations [26,27,28,29,30]. The presence of non-vaccine types, such as HPV66 and HPV35, highlights the need for continued local genotype surveillance to inform future vaccine updates and screening strategies. Based on the distribution observed in this cohort, the 9-valent vaccine would have covered 73.0% of HR-HPV infections, compared to 41.0% and 36.2% coverage from the quadrivalent and bivalent vaccines, respectively. These findings support calls for broader-valency vaccine strategies, especially in high-burden settings, where local HPV-type prevalence diverges from global patterns. Additionally, the WHO guidance recommends screening platforms capable of genotyping, as genotype-specific persistence is associated with differential risk for cervical precancer and can inform risk-based management pathways [20].

Associations were observed between HPV positivity and socio-demographic variables, specifically employment and marital status (*p* < 0.001), with higher prevalence among unemployed and single women. These findings align with regional studies linking socio-economic vulnerability to increased HPV acquisition and persistence [30,31]. In contrast, no significant associations were observed with age, ethnicity, province, or residence (urban vs. rural). The absence of age-specific trends may reflect the clinical context of this cohort, that is, women already seeking care and presumed sexually active, resulting in high prevalence across age groups. This observation could obscure age-specific patterns typically observed in general populations [32,33], thus emphasizing the importance of adapting cervical cancer screening strategies, including the selection of appropriate HPV test types and screening thresholds, to local epidemiological data [21].

While mRNA testing offers greater clinical relevance by detecting transcriptionally active infections [11], international guidelines, including those from the WHO [34] and the American Society for Colposcopy and Cervical Pathology (ASCCP) [35], recommend continued clinical monitoring for women with discordant results (i.e., HPV DNA-positive but mRNA-negative), especially those with prior cervical pathology or immunocompromise. Such considerations highlight the importance of integrating mRNA testing within broader clinical frameworks rather than as a standalone replacement.

Collectively, these findings emphasize the importance of assay characteristics in screening programs, especially in populations with a high HPV prevalence and cervical lesion burden. While DNA-based assays offer broader genotyping for surveillance, mRNA-based tests like APTIMA mRNA enhance clinical relevance by detecting viral oncogene expression associated with transformation. This approach aligns with the WHO recommendations to transition to HPV-based screening and highlights the value of biologically informed, operationally feasible tools in improving programmatic outcomes [20]. By focusing on identifying transcriptionally active, potentially persistent infections, mRNA platforms contribute meaningfully to evolving strategies aimed at enhancing the predictive value of cervical cancer screening.

### Limitations

This study has several limitations. While the sample size was adequate for assay comparison, it may not be representative of the broader South African population, limiting generalizability. The geographic focus may also not reflect regional variations in HPV prevalence or genotype distribution. The cross-sectional design restricts the assessment of infection persistence or progression, and the absence of histological endpoints (e.g., CIN2/3 lesions) limits the interpretation of the assays’ clinical accuracy. Additionally, cost-effectiveness was not evaluated, an important consideration for implementation in resource-limited settings. Despite these limitations, this study provides key insights into the comparative performance of DNA- and mRNA-based assays and contributes valuable data on genotype prevalence among South African women.

## 4. Materials and Methods

### 4.1. Study Design and Sample Size

A non-probability convenience sampling strategy was utilized, with participants enrolled until the target sample size was reached. The sample size was calculated using Epi Info version 7.1.5 (Centers for Disease Control and Prevention, Atlanta, GA, USA) to ensure 90% statistical power and 95% confidence, initially determined to be 416 but later increased to 527 to minimize potential study participant loss and information errors.

### 4.2. Study Sample and Sample Collection

This study sample consisted of 527 women aged 18 years and older who attended Termination of Pregnancy (TOP) and gynecology oncology clinics at the Dr George Mukhari Academic Hospital (DGMAH) in Ga-Rankuwa between January 2016 and December 2018. The women sought various gynecological services, including family planning, pregnancy termination, Pap smears requested by attending clinicians, colposcopy, large loop excisions of the transformation zone (LLETZ), and post-LLETZ reviews. Eligible participants had an intact cervix, provided informed consent, and completed a questionnaire. Women were excluded if they had a history of hysterectomy, were menstruating at the time of sampling, or had missing key data or compromised specimens. Participants were recruited from clinic waiting areas and enrolled after providing written informed consent. They then completed a questionnaire. Cervical samples (including endocervical, ectocervical, and transformation zone cells) were collected by healthcare workers using the Cervex-Brush^®^ Combi and a speculum (Rovers Medical Devices, Oss, The Netherlands). Each brush was immediately rinsed into a vial containing a 20 mL ThinPrep^®^ PreservCyt^®^ solution (Hologic, Inc., Marlborough, MA, USA) and then discarded. The vials were transported to the Virology Department at Sefako Makgatho Health Sciences University (SMU), where they were stored at room temperature until further processing and long-term storage and testing.

### 4.3. Sample Processing

Upon arrival at the department, 2 mL of each sample was aliquoted into 2 mL tubes for HPV testing, while the rest of the samples were used for liquid-based cytology testing.

### 4.4. DNA Extraction

DNA was extracted using the Abbott mSample Prep System DNA on the Abbott m2000sp platform (Abbott Molecular Diagnostics, Inc., Branchburg, NJ, USA). For each sample, 800 μL of PreservCyt cervical sample was aliquoted into labeled Abbott sample tubes and loaded onto the m2000sp. The extraction yielded 100 μL of DNA eluent per sample, which was transferred onto a 96 deep-well plate, then onto 96-PCR plates on the m2000sp for Abbott RealTime High Risk HPV testing. The extraction process included internal positive and negative controls to validate assay performance and monitor for contamination or technical errors.

### 4.5. HPV DNA Detection and Genotyping

#### 4.5.1. The Abbott RealTime High-Risk HPV Assay

The Abbott RealTime High-Risk HPV assay (Abbott Molecular GmbH & Co. KG, Wiesbaden, Germany), hereafter referred to as Abbott DNA, is a real-time PCR-based test designed to detect 14 HR-HPV types by targeting the conserved L1 gene of HPV. It individually identifies and reports results as HPV16 and HPV18, while the remaining 12 types (HPV 31, 33, 35, 39, 45, 51, 52, 56, 58, 59, 66, and 68) are collectively reported as “Other High Risk”. Amplification and genotyping were performed on the Abbott m2000rt system using 25 μL of extracted DNA per sample, in accordance with the manufacturer’s protocol. To ensure sample integrity and adequacy, the assay included an internal control that detects the endogenous human beta-globin sequence. Remaining eluates were aliquoted into labeled 2 mL microcentrifuge tubes and stored at −70 °C for subsequent testing with the Seegene Allplex™ HPV28 assay.

#### 4.5.2. The Allplex™ II HPV28 Detection Assay

The Allplex™ II HPV28 Detection assay (Seegene, Seoul, Republic of Korea), hereafter referred to as Seegene DNA, was performed in two tubes to permit the simultaneous amplification, detection, and differentiation of target nucleic acids for 19 HR HPV types (16, 18, 26, 31, 33, 35, 39, 45, 51, 52, 53, 56, 58, 59, 66, 68, 69, 73, and 82) and 9 LR HPV types (6, 11, 40, 42, 43, 44, 54, 61, and 70) following the manufacturer’s protocol. Five microliters of the extracted samples were used for HPV detection. The human beta-globin gene served as an internal control (IC) to monitor nucleic acid isolation and assess for potential PCR inhibition. For direct comparison with other assays, only the 14 HR-HPV types shared across platforms were included in the final analysis. The results are reported as positive or negative per genotype with cycle threshold (CT) values and IC data.

### 4.6. HPV E6/E7 mRNA Detection

HPV E6/E7 mRNA was tested using the APTIMA^®^ HPV assay on the PANTHER system (Hologic Gen-Probe, San Diego, CA, USA), hereafter referred to as APTIMA mRNA. One milliliter of the PreservCyt cervical sample was transferred into an APTIMA Specimen Transfer tube containing lysis solution and then tested. The APTIMA mRNA detects E6/E7 viral mRNA from 14 HR HPV types (HPV 16, 18, 31, 33, 35, 39, 45, 51, 52, 56, 58, 59, 66, and 68). The assay uses a non-infectious RNA transcript as an IC to monitor nucleic acid capture, amplification, and detection. The results are reported as positive or negative for mRNA, without specifying the individual HPV genotypes present.

### 4.7. Statistical Analysis

Statistical analysis was conducted using Stata version 18.5 (StataCorp, College Station, TX, USA). Assay performance was evaluated using sensitivity, specificity, and Cohen’s kappa (κ) statistic to assess agreement, with corresponding 95% confidence intervals (CIs). Agreement strength was interpreted as poor (κ ≤ 0.20), fair (κ = 0.21–0.40), moderate (κ = 0.41–0.60), good (κ = 0.61–0.80), or very good (κ = 0.81–1.00). Receiver operating characteristic (ROC) curve analysis was performed to assess discriminatory ability, using the Abbott DNA assay as the reference test. Area under the ROC curve (AUC) values were reported with 95% CIs. Prevalence ratios (PRs) with 95% CIs were calculated to compare detection rates across assays. A *p*-value < 0.05 was considered statistically significant.

## 5. Conclusions

This study highlights the comparative strengths and limitations of three HPV assays within a high-burden, resource-constrained setting. The Seegene DNA demonstrated the highest specificity and broadest genotype coverage, making it suitable for epidemiological surveillance. APTIMA mRNA provided comparable sensitivity to Abbott DNA and demonstrated analytical advantages by targeting transcriptionally active infections, supporting its clinical utility for risk stratification and more focused triage in cervical cancer screening programs.

The findings also show a genotype distribution dominated by HPV16, with notable prevalence of non-vaccine types, such as HPV66 and HPV35, reinforcing the need for ongoing surveillance and broader-valency vaccine strategies. Although limited by the absence of histological endpoints and longitudinal follow-up, this study’s findings emphasize the importance of aligning screening tools with local epidemiologic realities to optimize cervical cancer prevention efforts in South Africa and similar contexts. These findings support the WHO’s recommendation to transition to HPV-based molecular screening and underscore the potential programmatic value of mRNA assays, particularly in settings where over-referral and overtreatment may overwhelm limited health systems.

Although mRNA-based testing offers programmatic advantages by prioritizing clinically relevant infections, implementation as a primary screening tool should be approached cautiously. Complementary strategies, such as genotyping and follow-up protocols, remain essential to ensure comprehensive risk assessment. Consistent with the WHO’s guidance, it is important to acknowledge that women who are HPV DNA-positive but mRNA-negative may represent latent infections, and in some contexts, follow-up protocols may be warranted, particularly where additional risk factors are present. However, this study primarily focused on assay performance rather than long-term clinical outcomes. Overall, these results support the strategic adoption of molecular HPV testing aligned with the WHO’s screen-and-treat recommendations. They highlight the potential of scalable, context-adapted screening platforms, especially those that improve triage accuracy and resource utilization, toward reducing cervical cancer burden in South Africa and similar high-prevalence settings.

## Figures and Tables

**Figure 1 ijms-26-07450-f001:**
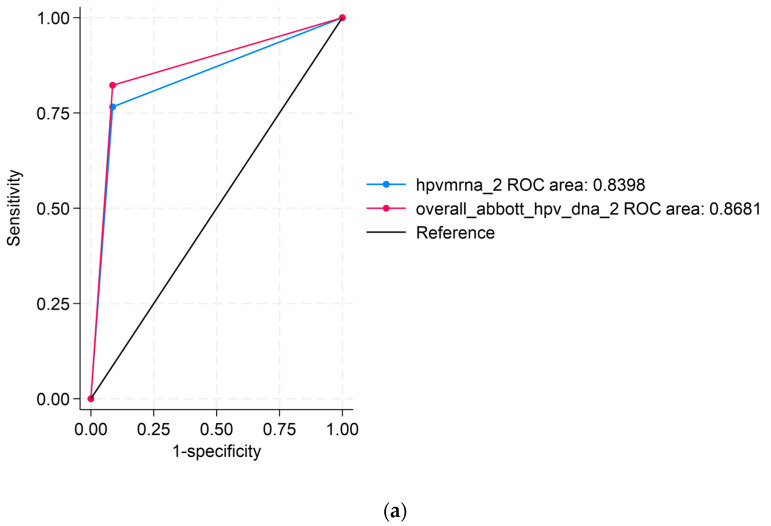
Receiver operating characteristic (ROC) curve analyses comparing HPV assays: (**a**) ROC curve comparing APTIMA mRNA to Abbott DNA, using Abbott DNA as the reference; (**b**) ROC curve comparing APTIMA mRNA to Seegene DNA. The black diagonal line in both panels represents the line of no discrimination (reference line), corresponding to an area under the curve (AUC) of 0.5. This line indicates the performance of a test with no diagnostic ability, where sensitivity equals the false positive rate. ROC curves that rise above this line indicate a better-than-random diagnostic performance.

**Figure 2 ijms-26-07450-f002:**
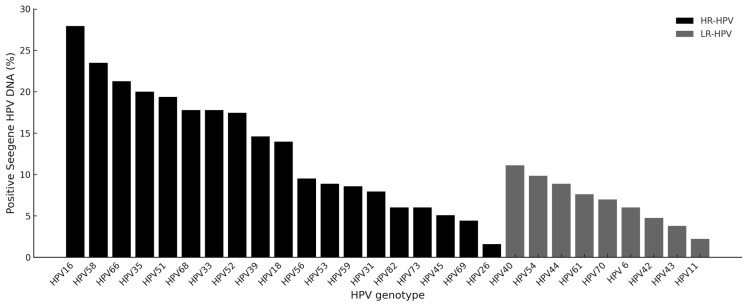
Overall prevalence (%) of all HPV genotypes (HR-HPV and LR-HPV) detected using the Seegene DNA.

**Figure 3 ijms-26-07450-f003:**
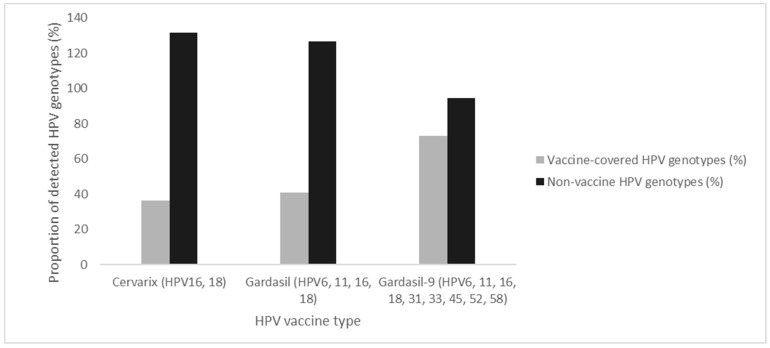
Distribution of HPV genotypes covered by licensed HPV vaccines.

**Table 1 ijms-26-07450-t001:** Performance comparison of Seegene DNA and APTIMA mRNA assays against the Abbott DNA assay.

(**a**) Seegene DNA vs Abbott DNA (14 shared genotypes)								
**Index Assay result**	**Abbott DNA Positive** ***n* (%)**	**Abbott DNA Negative** ***n* (%)**	**% Agreement**	**Sensitivity** **(95% CI)**	**Specificity** **(95% CI)**	**NPV** **(95% CI)**	**PPV** **(95% CI)**	**Cohen’s Kappa** **(95% CI)**
Seegene DNA Positive	232 (44.2)	50 (9.5)	82.9	82.3 (77.3–86.5)	91.4 (87.1–94.6)	81.6 (76.5–86.0)	91.7 (87.6–94.9)	0.7 (0.6–0.8)
Seegene DNA Negative	21 (4.0)	222 (42.3)						
Total	253 (48.2)	272 (51.8)						
(**b**) APTIMA mRNA vs. Abbott DNA								
APTIMA mRNA Positive	214 (40.6)	24 (4.6)	87.9	89.9 (85.4–93.4)	86.2 (81.6–89.9)	91.2 (87.2–94.3)	84.3 (79.2–88.5)	0.8 (0.7–0.8)
APTIMA mRNA Negative	40 (7.6)	249 (47.2)						
Total	254 (48.2)	273 (51.8)						

**Table 2 ijms-26-07450-t002:** Performance comparison of Abbott DNA and APTIMA mRNA assays against the Seegene DNA assay.

(**a**) Abbott DNA vs. Seegene DNA (14 shared genotypes)								
**Index Assay result**	**Seegene DNA Positive** ***n* (%)**	**Seegene DNA Negative** ***n* (%)**	**% Agreement**	**Sensitivity** **(95% CI)**	**Specificity** **(95% CI)**	**NPV** **(95% CI)**	**PPV** **(95% CI)**	**Cohen’s Kappa** **(95% CI)**
**Abbott DNA** Positive	232 (44.2)	21 (4.0)	86.5	91.7 (87.6–94.8)	81.6 (76.5–86.0)	91.4 (87.1–94.6)	82.3 (77.3–86.5)	0.7 (0.6–0.8)
**Abbott DNA** Negative	50 (9.5)	222 (42.3)						
Total	282 (53.7)	243 (46.3)						
(**b**) APTIMA mRNA vs Seegene DNA (14 shared genotypes)								
**APTIMA mRNA** Positive	216 (41.1)	21 (4.0)	83.4	91.1 (86.8–94.4)	77.1 (71.8–81.8)	91.4 (87.1–94.6)	76.6 (71.2–81.4)	0.7 (0.6–0.7)
**APTIMA mRNA** Negative	66 (12.6)	222 (42.3)						
Total	282 (53.7)	243 (46.3)						

**Table 3 ijms-26-07450-t003:** Performance comparison of Abbott DNA and Seegene DNA against the APTIMA mRNA.

(**a**) Abbott DNA vs. APTIMA mRNA								
**Index Assay result**	**APTIMA mRNA Positive** ***n* (%)**	**APTIMA mRNA Negative** ***n* (%)**	**% Agreement**	**Sensitivity** **(95% CI)**	**Specificity** **(95% CI)**	**NPV** **(95% CI)**	**PPV** **(95% CI)**	**Cohen’s Kappa** **(95% CI)**
Abbott DNA Positive	214 (40.6)	40 (7.6)	87.9	84.3 (79.2–88.5)	91.2 (87.2–94.3)	86.2 (81.6–89.9)	89.9 (85.4–93.4)	0.8 (0.7–0.8)
Abbott DNA Negative	24 (4.6)	249 (47.2)						
Total	238 (45.2)	289 (54.8)						
(**b**) Seegene DNA vs. APTIMA mRNA (14 shared genotypes)								
Seegene DNA Positive	216 (41.1)	66 (12.6)	83.4	76.6 (71.2–81.4)	91.4 (87.1–94.6)	77.1 (71.8–81.8)	91.1 (86.8–94.4)	0.7 (0.6–0.7)
Seegene DNA Negative	21 (4.0)	222 (42.3)						
Total	237 (45.2)	288 (54.8)						

**Table 4 ijms-26-07450-t004:** Comparative detection of 14 high-risk HPV types across molecular assays.

(a) APTIMA mRNA vs. Abbott DNA			
**APTIMA mRNA**	**Abbott DNA, *n* (%)**		**Total, *n* (%)**
	**Positive**	**Negative**	
**Positive**	213 (40.6)	24 (4.6)	237 (45.1)
**Negative**	40 (7.6)	248 (47.2)	288 (54.9)
**Total**	253 (48.2)	272 (51.8)	525 (100.0)
(b) APTIMA mRNA vs. Seegene DNA			
**APTIMA mRNA**	**Seegene DNA, *n* (%)**		**Total, *n* (%)**
	**Positive**	**Negative**	
**Positive**	216 (41.1)	21 (4.0)	237 (45.1)
**Negative**	66 (12.6)	222 (42.3)	288 (54.9)
**Total**	282 (53.7)	243 (46.3)	525 (100.0)
(c) Abbott DNA vs. Seegene DNA			
**Abbott DNA**	**Seegene DNA, *n* (%)**		**Total, *n* (%)**
	**Positive**	**Negative**	
**Positive**	232 (44.2)	21 (4.0)	253 (48.2)
**Negative**	50 (9.5)	222 (42.3)	272 (51.8)
**Total**	282 (53.7)	243 (46.3)	525 (100.0)

**Table 5 ijms-26-07450-t005:** Socio-demographic correlates of HPV positivity by assay type.

Socio-Demographic Variables	*n* (%)	Positive HPV Infections					
		Abbott DNA (N = 527), *n* (%)	*p*-Value	Seegene DNA (N = 525), *n* (%)	*p*-Value	APTIMA mRNA (N = 527), *n* (%)	*p*-Value
**Age (years)**			0.090		0.253		0.379
<30	151 (28.8%)	86 (16.3%)		100 (19.1%)		71 (13.5%)	
30–39	173 (33.0%)	78 (14.8%)		99 (18.9%)		71 (13.5%)	
40–49	120 (22.9%)	59 (11.2%)		74 (14.1%)		62 (11.8%)	
50–59	60 (11.5%)	24 (4.6%)		33 (6.3%)		26 (4.9%)	
60–68	19 (3.6%)	7 (36.8%)		9 (1.7%)		7 (1.3%)	
Total	524 (100.0%)	254 (48.2%)		315 (60.0%)		237 (45.0%)	
**Ethnicity/Race**			0.299		0.414		0.270
Black	526 (99.8%)	253 (48.0%)		314 (59.8%)		237 (45.0%)	
Colored	1 (0.2%)	1 (0.2%)		1 (0.2%)		1 (0.2%)	
Total	527 (100.0%)	254 (48.2%)		315 (60.0%)		238 (45.2%)	
**Province**			0.808		0.490		0.812
Gauteng	470 (90.2%)	230 (43.6%)		285 (54.3%)		214 (40.6%)	
Limpopo	2 (0.4%)	1 (0.2%)		1 (0.2%)		1 (0.2%)	
Mpumalanga	1 (0.2%)	0 (0.0%)		0 (0.0%)		0 (0.0%)	
North West	48 (9.2%)	23 (4.4%)		26 (5.0%)		23 (4.4%)	
Total	521 (100.0%)	254 (48.2%)		312 (59.4%)		238 (45.2%)	
**Location (Urban/Rural)**			0.883		0.883		0.507
Urban	9 (1.7%)	5 (1.0%)		6 (1.1%)		2 (0.4%)	
Semi-Urban	452 (86.8%)	220 (41.8%)		270 (51.4%)		208 (39.5%)	
Semi-Rural	43 (8.3%)	22 (4.2%)		27 (5.1%)		21 (4.0%)	
Rural	17 (3.3%)	7 (1.3%)		9 (1.7%)		7 (1.3%)	
Total	521 (100.0%)	521 (98.9%)		312 (59.4%)		238 (45.2%)	
**Employment status**			<0.001		<0.001		<0.001
Employed	214 (43.2%)	79 (15.0%)		110 (21.0%)		75 (14.2%)	
Unemployed	283 (56.9%)	166 (31.5%)		191 (36.4%)		153 (29.0%)	
Total	497 (100.0%)	497 (93.3%)		301 (57.3%)		228 (43.3%)	
**Marital/Relationship status**			<0.001		<0.001		<0.001
Married	116 (23.3%)	37 (7.0%)		50 (9.5%)		34 (6.5%)	
Single	337 (67.8%)	189 (35.9%)		224 (46.7%)		174 (33.0%)	
Divorced/ Separated	44 (8.9%)	19 (3.6%)		27 (5.1%)		20 (3.8%)	
Total	497 (100.0%)	497 (93.3%)		301 (57.3%)		230 (43.6%)	

## Data Availability

The data presented in this study are available on request from the corresponding authors.

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
