# Peer review of "APTIMA mRNA vs. DNA-Based HPV Assays: Analytical Performance Insights from a Resource-Limited South African Setting"

_ijms, 2025, doi:10.3390/ijms26157450_

Round 1

Reviewer 1 Report

Comments and Suggestions for Authors

The authors present a well written evaluation of the APTIMA mRNA HPV assay comparing its analytical performance to well known DNA assays, in particular genotyping assays. The resutls are presented well and the comparisons of specificity and sensitivity are sound. I did appreciate the section on limitations of the study. It conveyed that the authors had considered the practical and technical aspects, specifically with respect to the different molecular formats of the assays.

The manuscript would be enhanced if the authors can include a discussion about how using the mRNA test could improve resource use for treatments. While the motivation for this study is clear, it is not obvious exactly what kind of treatment would be impacted by implementing mRNA testing. Perhaps provide a case study. This will underscore the relevance of the study.

Reviewer 2 Report

Comments and Suggestions for Authors

The article: ijms-3732292

     Dear Ms. Tansy Wang

The study "Superior Analytical Specificity of APTIMA mRNA HPV Testing Over DNA Assays in a Resource-Limited South African Setting" holds relevance in the diagnostic investigation and follow-up of gynecological cancer. Although the Aptima® platform has been previously described in the literature, its application in the context of diagnostic investigation among women in South Africa is of particular importance, considering the epidemiological characteristics and the specificities of healthcare pathways in different regions.

Tests for the detection of viral messenger RNA (mRNA) are highly relevant in diagnostics, as they reflect the transcriptional activity of the virus in infected cells, indicating active infection. The identification of HPV mRNA in samples previously positive for viral DNA represents a significant advancement in molecular diagnostics aimed at gynecological cancer. It enables the optimization of complementary testing workflows and supports a more rational clinical approach, reducing unnecessary referrals for colposcopy.

The Aptima® automated platform is designed to detect the mRNA of the E6/E7 oncogenes from the 14 most prevalent high-risk HPV genotypes (HPV 16, 18, 31, 33, 35, 39, 45, 51, 52, 56, 58, 59, 66, and 68), and does not include other genotypes also classified as high-risk, such as HPV 26, 53, 73, and 82. This limitation may restrict the diagnostic scope of the methodology in certain epidemiological settings, especially in populations where these less common genotypes circulate more frequently.

Nevertheless, its application remains indisputably relevant within the diagnostic pathway of HPV infection, as it covers the genotypes with the greatest clinical and epidemiological impact. However, it is essential to emphasize the importance of appropriate follow-up for cases that are mRNA-negative but DNA-positive, as these patients may be in a latent phase of infection or at potential risk of progression, requiring continuous clinical and laboratory monitoring.

International guidelines, such as those from the American Society for Colposcopy and Cervical Pathology (ASCCP) and the World Health Organization (WHO), recognize the importance of clinical and laboratory follow-up for women with high-risk HPV DNA, even in the absence of abnormal cytology or detectable mRNA. Follow-up should always take into account the patient’s age, prior history of cervical lesions, and immunological status.

   The attached article includes comments on aspects that should be incorporated into the text presented to the authors.
Finally, the article should be accepted after the suggested considerations have been addressed.

Kind Regards
